# Design and Application of Deep Hash Embedding Algorithm with Fusion Entity Attribute Information

**DOI:** 10.3390/e25020361

**Published:** 2023-02-15

**Authors:** Xiaoli Huang, Haibo Chen, Zheng Zhang

**Affiliations:** Research Institution of Signal Detection and Information Processing Technology, Xihua University, Chengdu 610039, China

**Keywords:** hash embedding, deep learning, attribute information, entity coding

## Abstract

Hash is one of the most widely used methods for computing efficiency and storage efficiency. With the development of deep learning, the deep hash method shows more advantages than traditional methods. This paper proposes a method to convert entities with attribute information into embedded vectors (FPHD). The design uses the hash method to quickly extract entity features, and uses a deep neural network to learn the implicit association between entity features. This design solves two main problems in large-scale dynamic data addition: (1) The linear growth of the size of the embedded vector table and the size of the vocabulary table leads to huge memory consumption. (2) It is difficult to deal with the problem of adding new entities to the retraining model. Finally, taking the movie data as an example, this paper introduces the encoding method and the specific algorithm flow in detail, and realizes the effect of rapid reuse of dynamic addition data model. Compared with three existing embedding algorithms that can fuse entity attribute information, the deep hash embedding algorithm proposed in this paper has significantly improved in time complexity and space complexity.

## 1. Introduction

With the popularization of the Internet, a large amount of text data is generated every day, and the automatic analysis of data and information extraction technology are expected to be applied in various situations. Generally, machine learning and deep learning can only be learned and calculated in the form of mathematical vectors, and the field of practical application contains a large number of discrete and character-type discontinuous digital features. As an important feature extraction algorithm, the embedding algorithm [1] converts high-dimensional sparse features into low-dimensional dense features, or maps discrete variables into low-dimensional continuous spaces, which plays an important role in both machine learning and deep learning. This is usually the first step of various deep learning tasks. The expression effect and embedding performance often determine the results of subsequent tasks, such as the recommendation accuracy of the recommendation model and the understanding of semantics in the NLP model.

In general, the extremely popular one-hot technique is expressed by assigning a unique index to a single word. While this approach is very simple and easy to understand, it assumes that each word is independent, so synonyms and similar words are often treated as unrelated words. Furthermore, the problem with one dimension being assigned to one word is that, as the vocabulary grows, it leads to dimension explosion, which is difficult to handle. Therefore, researchers have studied a large number of methods to solve this problem. Word2Vec [2,3], proposed by Micolov et al., can generate word vectors with multi-dimensional representations through unsupervised learning from a large corpus, where words with similar meanings have similar weights. Word2Vec expresses the meaning of words by computing word vectors and makes it possible to deal with the relationship between words. Since then, embedding technology has developed rapidly in the fields of natural language processing, recommendation algorithms and graph data science.

In 2016, Facebook proposed the fastText [4,5] algorithm, which can greatly reduce the training time. Referring to the Word2vec method, Microsoft proposed the item2vec [6,7] algorithm, which regards the products in the e-commerce platform as a vocabulary expressed in the question, and regards the collection of all products as an article, so as to derive the item2vec algorithm. The abstract ability of the model can find the hidden factors, which well solves the problem that it is difficult to find the influencing factors between items and users. Youtube [8] uses a neural network to combine various basic information of users and train them as user vectors. In the entire recommendation model, the SoftMax weight is output as video embedding. In the same year, Google proposed the Wide&Deep [9] model. The wide part provides interpretable properties, and the deep part is used to discover high-order combination features not found in the training set. In 2020, Google proposed the DHE [10] model to effectively solve the problems of dynamic data increase, excessive data volume occupying too much memory, and uneven data distribution.

In the field of graph embedding, the DeepWalk algorithm [11], proposed in 2014, is a representative common algorithm for graph embedding, which is used to learn the vector representation of vertices in the network for further machine learning. The idea of DeepWalk is similar to Word2Vec, a commonly used algorithm in the NLP field, which uses the co-occurrence relationship between nodes and nodes in the graph to learn the vector representation of nodes. It samples nodes in the graph by means of random walks to simulate the corpus in the corpus and learn the co-occurrence relationship of nodes, so that similar nodes in the original graph are also close in their low-dimensional expression space. In 2017, Stanford University proposed expanding the GraphSAGE algorithm [12] into an inductive learning task by training the function of aggregating node neighbors, which could quickly generalize unknown nodes to obtain embedding vectors for evolving graph structures. The generative embedding of unknown data is achieved by learning an embedding function, sampling and aggregating features using local neighborhoods of nodes. The FastRP algorithm [13,14], proposed by Chen et al., in 2019, allows iterative calculation of node embedding, maintains the similarity between nodes and neighbor nodes, and does not need to explicitly construct a similarity matrix to further improve the speed. Among the algorithms using hash [15,16,17] ideas, Hash2Vec proposes applying feature hashing to create word embeddings for natural language processing. This algorithm does not require training to capture the semantics of words and can easily process a huge vocabulary of millions of orders of magnitude. Binary-oriented hash embedding [18] learns to arbitrarily compress storage space while maintaining model accuracy.

There have been many research results in the embedding field of word embedding [19], graph embedding [20,21] and item embedding. However, most embeddings are based on the contextual relationship between entities, and do not integrate multiple feature attributes within entities. This paper designs a deep hash embedding (hereinafter referred to as FPDH) algorithm that fuses entity attribute information. The time and space consumption under different embedding dimensions are tested, the clustering effect is tested through the distribution in space, and the mutual influence and characteristics of entities in their vector space are analyzed.

## 2. Introduction to Related Algorithms

The descriptions of the symbols used in the text are shown in Table 1.

### 2.1. The Introduction of One-Hot Encoding

Data mining and storage usually use data in the form of natural language text, which makes it much easier for humans to read and understand. However, most machine learning algorithm models cannot be used directly. The solution is to convert text data into numeric data. For example, “sun”, “moon”, “earth”, become 0 for the Sun, 1 for the Moon, and 2 for the Earth. The category values in the form of a set containing a set field of *n* categories can be easily viewed as a continuous numeric type with constant integers in the interval [0, *n* − 1]. One-hot coding can effectively solve this problem. One-hot encoding is for a set of *n* categories, with n-bit state flag bits to encode each class. Each flag bit corresponds to one class, and only one flag bit is one at any time, while the rest is 0. Defining the encoding function *E* with one-hot encoding *E*(*s*) for the feature word *s* is defined as Formula (1):(1)$$\left\{\begin{array}{c} E\left(s\right)=b\in {\left\{0,1\right\}}^{n}\\ b_{s}=1,b_{j}=0\left(j\neq s\right) \end{array}\right.$$

For example, “sun”, “moon”, “earth” are respectively depicted as shown in Figure 1:

The codes of these three categories of feature words are [1, 0, 0], [0, 1, 0], and [0, 0, 1]. If there are n categories in the set domain, the one-hot coding is a vector of length n. In each case, only 1 component of each vector is 1 and the others are 0. This determines the uniqueness of each category.

One-hot encoding is performed for each feature word with a set domain to obtain a vector representation of all feature words, which is essentially an embedding table (W $\in R^{n*d}$) composed of all feature word vectors. By querying the s-th row of the embedding table W [22], we obtain the representation vector e. One-hot encoding is essentially a neural network that can be viewed as a layer of unbiased terms. The corresponding embedding is expressed as $e=W^{T}b$. After processing data using one-hot coding for discrete data, downstream tasks, such as similarity calculation and distance calculation, are more useful, applicability to related machine models is more appropriate, and availability is improved [23,24].

### 2.2. The Introduction of Hash Embedding Algorithms

Hash, also known as hashing, converts an input of arbitrary length into an output of fixed length by a hashing algorithm, where the output value is the hash value. If a record in the hash table corresponds to the original input v of the algorithm, then v must be at the position of H(v). If the hash values of the two feature words are different, then the original inputs, corresponding to the two hash values, must be different. For large datasets, the vocabulary size can reach hundreds of thousands, and the model parameters can easily reach millions, or even hundreds of millions. With a large number of vocabularies, there are problems with the size of the embedding. Due to the excellent properties of the hash function, all these problems can be solved by hash embedding.

The use of hash embedding has the following four advantages: (1) When using hash functions, it is not necessary to implement the creation of a dictionary that can handle dynamically growing vocabularies; (2) Hash embedding has a mechanism that allows implicit vocabulary cleaning; (3) Hash embedding is implemented based on hash. This method adds a trainable mechanism and effectively solves hash conflicts. (4) Hash embedding usually reduces the parameters by several orders of magnitude.

Hash functions inevitably have their own shortcomings. For a large amount of data, using a hash map for a large number of original inputs leads to hash value collisions, i.e., multiple initial values with the same hash value. When the hash value directly represents the original input, it is impossible to distinguish which initial input it is, affecting the model’s impact. Hash embedding uses several different hash functions to represent the same initial value to reduce the possibility of collisions. The overall schematic representation of the algorithm can be seen in Figure 2.

Steps of the algorithm:

Step 1: Use *k* different hash functions $H_{1},\ldots \ldots ,H_{k}$ for the feature word *s* to be embedded and calculate *k* different hash values.

Step 2: Combine the *k* components from Step 1 into a weighted sum:(2)$$\hat{e_{s}}=\sum_{i=1}^{k}p_{s}^{i}H_{i}\left(s\right),\;p_{s}={\left(p_{s}^{1},\ldots \ldots ,p_{s}^{k}\right)}^{T}\in R^{k}$$

Step 3: Optional. The weighting parameters $p_{y}$ of the feature words can be concatenated with ${\hat{e}}_{s}$ to pro *e* the final vector *e*.

The feature word *s* is converted into a complete representation of the hash code (⊕ is the concatenation operator) according to Formula (3)
(3)$$\left\{\begin{array}{c} c_{s}={\left(H_{1}\left(s\right),\ldots \ldots ,H_{k}\left(s\right)\right)}^{T}\\ p_{s}={\left(p_{s}^{1},\ldots \ldots ,p_{s}^{k}\right)}^{T}\\ \hat{e_{s}}=p_{s}{}^{T}c_{s}\\ e_{s}^{T}={\hat{e}}_{s}^{T}⊕p_{s}^{T}\;\left(\mathrm{optional}\right) \end{array}\right.$$

If step 3 is not selected, ${\hat{e}}_{s}$ is the final vector representation, and if step 3 is selected, $e_{S}^{T}$ is the final vector representation.

## 3. Embedding Algorithm Based on a Deep Hash Fusion of Entity Information

As the amount of data increases, and the content becomes richer, the entity description information becomes more comprehensive. Embedding algorithms are an area with extremely high requirements for expressive features that are both dynamic and highly complex. In practical applications, entities that belong to a particular category have category–attribute labels in the same domain. For example, a movie can be considered an entity, and its attribute labels can be divided into director, actor, genre, language, rating, etc. A car can also be considered an entity, and its attribute labels can be divided into performance type, stage, engine capacity, power, color, etc. In this paper, we develop a deep hash embedding model that combines information about entity attributes to improve representation. The overall structure of the model is shown in Figure 3.

The model structure is divided into three parts:Entity feature encoding: Firstly, the first part fuses and encodes all the features of the entity. The purpose is to express the features of additional information in the vector representation of the entity node. After encoding, a concatenated encoding of variable length $v_{\mathrm{initial}\;\mathrm{encoding}}$ is obtained.Feature extraction after encoding: In the second part of feature extraction, the variable length initial encoding of v is converted into k fixed hash values through k non-repetitive hash functions, and different entities can already be represented at this time.Using neural networks for feature learning [25]: In the third part, we train a general deep neural network that inputs k different hash values encoded by each entity separately into the neural network, learns the inherent, implicit association between k features, and outputs the outputs in the desired dimension to obtain the embedding vector.

### 3.1. Entity Feature Encoding

This paper proposes a new encoding method, namely, Domain feature-based entity encoding. First, we define E as the set of all domains of the entity, according to all attribute information in the entity.

A new encoding method is proposed wherein encoding entities are based on domain attributes. First, after all attribute information is in a class of entities, we count all attribute categories belonging to |E| different fields and define E as the set of all fields, $E=\left\{F_{1},\ldots \ldots ,F_{n}\right\}$, e.g., movie entity,
(4)E={Actor,Director,Label,Country,Language}

|E| indicates the number of sets E. Each domain $F_{i}$ contains m different features $F_{i}=\left\{f_{1},\ldots \ldots ,f_{m}\right\}$, and each feature $f_{i}$ belongs to only one domain $F_{i}$. The overall flow of the encoding algorithm is shown in Figure 4:

Step 1: Count all domains E and the number of domains |E| contained in a class of entity records.

Step 2: Count all the features F contained in each range, sorting them so that the order of F is fixed each time, and count the number of all features |F|.

Step 3: If the features in the domain are discrete variables, set a vector V of length |F| containing all zeros for this domain. Continuous variables are not processed in this process for now and are processed in the following 2.3 Feature Learning.

Step 4: Further improvement is based on one-hot coding. One coding for each tag isno longer used, but rather one coding is used for a range. Set the encoding for the entity in a particular domain. Determine the domain $F_{i}$ to which each feature $f_{i}$ of the entity belongs, find the position index of $f_{i}$ in $F_{i}$, and set the value of the index in the vector V to change from 0 to 1.
(5)$$V={\left\{0,1\right\}}^{|F|}$$
(6)$$\left\{\begin{array}{c} \;\mathrm{if}:\;F\left[\;\mathrm{index}\;\right]=f_{i},V\left[\;\mathrm{index}\;\right]=1\\ \;\mathrm{if}:\;F\left[\;\mathrm{index}\;\right]\neq f_{i},V\left[\;\mathrm{index}\;\right]=0 \end{array}\right.$$

Step 5: Perform step 4 coding for each domain in the entity, and code to get the vector $V_{i}$ of each domain.
(7)$$e^{\prime }=V_{1}∥V_{2}∥\cdots ∥V_{n}$$

Taking the movie entity as an example of coding, Figure 5 is an attribute domain diagram of the attribute information contained in the movie, and Figure 6 shows the attribute information in each domain of The Shawshank Redemption.

Throughout the movie dataset, all attribute information includes eight categories: Director, Year, Rating, Actor, Label, Country, Genre, and Language. The discrete data types are actor, director, label, country, and language. There are five domains.



E={Actor,Director,Label,Country,Language}



First, the number of all features in each field is counted, and, for each field, a vector of zeros is set, the length of which is equal to the number of features in the field.

Sort all features so they are ordered and fixed in sequence, with each bit representing a feature. If that feature occurs in the movie, set the 0 representing the job to 1. The initial code s is obtained by concatenating the representation vectors of all domains. A detailed coding example is shown in Figure 7.

### 3.2. Extract Features Using Hash Functions

After the attribute information is merged, the initial encoding of the entity is the sum of the lengths of all fields, and the initial encoding is represented by a string composed of 0 and 1. Although the hash method can effectively reduce the length of the code, in the case of an ideal uniform distribution, the number of entities in the data set n, on average, has n codes mapped to a smaller space m at the same time, and the formula is as follows: (8)H(x)=((ax+b)modp)modm

*a* and *b* are random integers, b≠0 and *p* is a large prime, much larger than *m*.

Usually, m is much smaller than n in the hash algorithm, which usually causes collision problems. There may be two or more entity-encoded values represented by the same hash value. Different entities have the same vector representation, which inevitably affects the performance of the model, and the model is not be able to distinguish different feature values. In order to alleviate the conflict problem, multiple hash functions are used to extract features to jointly represent an entity. The core idea is to connect multiple different hash functions in series, and the final result greatly reduces the possibility of conflict.

The vector encoded by the initial part is encoded with several hash functions, as shown in Figure 8, using encoding function $E_{k}\to H_{k}$. The eigenvalues are mapped to k dimensions using K general hash functions. $E^{\prime }\left(s\right)=\left[H^{1}\left(s\right),\ldots \ldots ,H^{k}\left(s\right)\right]$ Where H:V→[m], m has nothing to do with the size of the embedding table, just set m to a larger number, then the hash function can hash evenly on 1, 2, 3, …, m on. The hashed eigenvalues become k-dimensional features, but the integer $E^{\prime }\left(s\right)$ is unsuitable input for the neural network. Therefore, the encoding must be done by a suitable transformation.
(9)$$\left.E\left(s\right)=\mathrm{Transform}(E^{\prime }\left(s\right)\right)$$

The transformation function chooses a uniform distribution for simplicity and ease of operation.
(10)$$\mathrm{Transform}\left(x\right)=2\left(\frac{x-1}{\;m-1}\right)-1$$

No memory is required during this encoding process since all computations can be performed dynamically. This is also a nice feature of using multiple hashes, as we get cleaner high-dimensional encoding without increasing the amount of memory required.

### 3.3. Feature Learning

The third part uses the deep neural network to learn the internal hidden features, integrates various attribute information of the entity, and learns the inherent implicit association of the entity through the neural network with strong learning ability.

Although the k-dimensional integer features after hash processing in 2.2 can already distinguish different entities, they are not suitable for direct retrieval and query, and it is difficult to apply them to downstream tasks, which require mapping and conversion of data. The process is very similar to highly nonlinear feature transformations, where the input features interact and are not easy to learn. Deep neural network is a general function approximator, so using deep neural network models to fit such complex transformations is very suitable.

The attribute information of the entity includes discrete variables and continuous variables. In step 2, only the area of the entity belonging to the discrete variables is encoded. Some of the constant variables only need to perform simple normalization to use the neural network. The network learns features and output features.

Simple linear normalization transforms continuous data into [0, 1]. The formula is as follows:(11)$$X_{\mathrm{norm}\;}=\frac{X-X_{\min}}{X_{\max}-X_{\min}}$$

This method realizes the proportional scaling of the original data, such that *X* is the original data, $X_{\max}$ is the maximum value of the original data set, and $X_{\min}$ is the minimum value of the original data set.

After the year and score in the movie entity attributes are normalized by the formula, they are input to the feedback neural network, together with the k-dimensional features extracted in step 2, to learn the intrinsic features, as shown in Figure 9.

Set an all-1 vector of length |e| as a control, and use Jaccard similarity to calculate the similarity values as the label value of the feedback neural network as supervision. This process uses a single hidden layer feedforward neural network. The input data vector is ${\left(x_{1},x_{2},\ldots ,x_{n}\right)}^{T}$ and the output y is data. Taking the network structure where the hidden layer is set to 1 layer as an example, the principle of multi-layer hidden layers is the same. The input and output of the hidden layer nodes are as follows: (12)$$n_{ji}=\sum_{i=1}^{n}\omega_{ji}\times x_{n-i}+a_{i}$$
(13)$$x_{j}=f_{j}\left(\;\mathrm{net}{\;}_{\mathrm{ji}}\right)$$

The input and output of the output layer node are, respectively, the formula: (14)$$n^{net}{}_{jo}=\sum_{i=1}^{n}\omega_{oj}\times x_{n-i}+a_{o}$$
(15)$$x_{j}=f_{o}\left(\;\mathrm{net}{\;}_{oj}\right)$$
$\omega_{ji}$, $\omega_{oj}$ is the weights of the connection between the input layer and the hidden layer and the weights of the connection between the hidden layer and the output layer are in the range [−1, 1]. $f_{j},f_{0}$ are the activation functions of the hidden layer and the output layer, respectively.

During the backpropagation process of the BP algorithm, when there is a difference between the actual output ŷt and the expected output $y_{t}$, the error calculation is as follows: (16)$$E=\frac{1}{2}{\left(\hat{y_{t}}-y_{t}\right)}^{2}$$

According to the gradient descent method, the update formula of the weights can be obtained as: (17)$$\Delta \omega_{ed}=-\lambda \frac{\partial E}{\partial \omega_{ed}}$$
where λ is the learning rate and $\omega_{ed}$ is the connection weight between two nodes. After the model is trained, the eigenvalues of the entity are input to the neural network for calculation, and the d-dimensional output of the calculation result of the last hidden layer (the blue part in Figure 8) is used as the final embedding vector.

## 4. Experimental Simulation and Analysis

### 4.1. Experimental Setup and Dataset

The experiment used a computer with Intel(R) Core(TM) i5-8257U CPU @ 1.40GHz and 16g memory.The operating system was macOS, the compiler software was python3.8, and the deep learning framework was pytorch1.10.0. To verify the proposed FADH algorithm, experiments were carried out on the same data set using different angles. Table 2 gives the detailed statistics of the data set.You can find the experiment code in https://github.com/Minglechb/kg_dhe_rec, accessed on 28 November 2022.

The dataset’s source, hetrec2011-movielens-2k-v2, was the 2nd International Symposium on Information Heterogeneity and Fusion for Recommender Systems, an extended MovieLens-10m, including 10,197 movies, 20 movie types, 4060 directors, and 95,321 actors, each film has an average of 22.778 actors, the number of countries in the movie was 10,197, and each movie had an average of 1 attribution country, involving a total of 47,899 attribution country locations.

### 4.2. Baseline Model

The baseline method is as follows:

Node2vec: Node2vec [26] defines the concept of a flexible vertex network domain and implements a biased random walk technique that explores different neighborhoods to achieve richer representations.

GraphSage: GraphSage is an inductive learning framework that combines topology and entity vertex attribute information to efficiently generate embeddings of unknown vertices in graphs. The core idea is to generate the embedding vector of the target vertex by performing an aggregation representation function on a pair of neighbor vertices.

FastRP: Fast Random Projection is a node embedding algorithm in the random projection algorithm family. The Johnson–Lindenstrauss lemma theoretically supports these algorithms, according to which n vectors of arbitrary dimension can be projected to O(log(n)) dimension. Such techniques allow aggressive dimensionality reduction while preserving most of the distance information. The FastRP algorithm operates on graphs and, in this case, is concerned with maintaining the similarity between nodes and their neighbors. This means that two nodes with similar neighborhoods should be assigned similar embedding vectors. Conversely, two dissimilar nodes should not be assigned similar embedding vectors.

### 4.3. Parameter Setting


FADH Model parameter settingsFor all datasets, the size of the embedded entity dimension was [32, 64, 128, 256, 1024] for the baseline algorithm model and the parameter model of the FADH algorithm in this paper. The number of training rounds was 1, and the initial learning rate of the neural network was 0.01. The learning rate was reduced by a minimum of 0.0001 after each training iteration. The network structure designed in this paper is shown in Table 3 below:Since the equal-width deep network had higher parameter utilization, the model effect was better. In this experiment, the neural network model was set as the input layer of 1024 dimensions, the middle hidden layer had six layers, the first three layers were all 1024 nodes, and the sixth hidden layer had the same number of nodes as the required dimension.Baseline model parameter settingsThe Node2vec model parameters were set. The depth of the random walk was 80, the number of random walks generated by each node was 10, and the localization parameter (the tendency of the random walk to stay close to the starting node or fan out) was set to 1. A higher value meant keeping the local state. The training neural network context window size was 10. The number of negative samples generated for each positive sample was 5.GraphSage and FastRP model parameters were set according to the parameters shown in the literature.


### 4.4. Data Visualization

To further understand the performance results of the algorithm and observe the obtained vector space, the T-SNE dimensionality reduction algorithm was used. Entities with similar attribute features should be closer together in the space vector, as shown in the figure.

In the experimental data of this paper, each entity had attributes of 8 domains, encoded by the FADH algorithm and was then reduced to 2-dimensional details by T-SNE for visual analysis. An example of dimension data is shown in Table 4.

Show some nodes in two-dimensional space, as shown in Figure 10.

Figure 10 shows the distribution of entities in a two-dimensional space, which could intuitively see the spatial distance of entities and intuitively feel the similarity between entities.

At present, the application of embedding has been very extensive, but there is no perfect solution to the problem of embedding evaluation. In real engineering applications, the way to evaluate the quality of embedding is to judge by the impact index of the embedding vector on specific tasks. The most commonly used method is to calculate the TOP N similarity, check whether the distance of the entity in the space is consistent with human intuition, and check whether the model has learned meaningful information, or to learn from the clustering visualization method in word embedding, and compare and check the embedding as good or bad. The evaluation embedding scheme in this paper involved encoding the movie entity, calculating its relative distance, visualizing it with a clustering algorithm, and visually seeing the distribution effect of the vector to verify whether the embedded vector had learned the internal features.

In the process of clustering, the quality of clustering was determined by calculating the squared error (*SSE*) after the centroid converged, calculated as follows: (18)$$SSE=\sum_{i=1}^{n}{\left(y_{i}-y_{i}^{*}\right)}^{2}$$

The closer the *SSE* was to 0, the better the model fitting effect was. The k-means clustering algorithm was used to determine the number of clusters by the K-elbow method. As the clustering coefficient K increased, the sample division was more refined, the degree of aggregation of each cluster gradually increased, and the sum of squared errors gradually decreased. The results of clustering coefficient K and mean square error are shown in Figure 11.


When the clustering coefficient was from 1 to 5, the sum of squares of errors decreased rapidly, indicating that the number of clusters was quickly optimized to find the optimal number of clusters within this interval.When the clustering coefficient was 5 to 20, the rate of decline of the clustering coefficient slowed down, and the most likely interval of the number of clusters was most likely to be close to the actual situation.When the clustering coefficient was 20 to 100, as the clustering coefficient increased, the decrease of the sum of squared error was the slowest. The rise in the number of clusters exceeded the actual number of clusters represented by the data. The division of the number of clusters directly affected the performance of downstream.


The division of the number of clusters directly affected the performance of downstream tasks. When the number of classifications was too small, and the characteristics of the entities in the clusters were too different. The overall category could not effectively reflect the characteristics of the entities when the number of clusters was too large. If the distinction between clusters was too small, the significance of clustering was lost. In addition, too many clusters consumed a lot of computing resources. In the application of downstream tasks, the clustering coefficient was determined according to the requirements of the task. A few specific cases were compared here.

When the clustering coefficient was 5, the clustering effect is shown in Figure 12. At this time, the number of clusters began to approach the real situation. The figure shows that a large amount of data was regularly divided into 5 categories, but there was not yet sufficient distinction. Each class contained a large number of entities, and its category could not effectively reflect the characteristics of the entity. The entity characteristics within the category were quite different. In the downstream task, the category was used to replace an entity. In this case, the category could not effectively reflect the real situation, resulting in poor overall performance of the downstream task.

When the clustering coefficient was 100, the clustering effect diagram is shown in Figure 13. At this time, the error was small, the resource consumption was large, the calculation time was long, the number of individuals in the category was greatly reduced, and there were only a few in each category. There were too many types of entities used in downstream applications to significantly reduce computing resources.

After considering a reasonable trade-off between the error and the number of clusters, a clustering coefficient having an error that no longer decreased significantly and a cluster in which the number of individuals was moderate was selected. The clustering category could be used instead of the division of individuals, and the number of clusters was finally determined to be 20. It was also the same as the number of categories of movies, as shown in the Figure 14.

Using the K-elbow method to select an appropriate clustering coefficient, categories could be used instead of individuals in downstream tasks. After the FPDH embedding model was integrated into the attribute characteristics of entities, it learnt the potential association between feature attributes and effectively distinguished individuals with similar characteristics.

### 4.5. Algorithm Performance

A single feature was calculated by relative clustering distance.

The primary method of a single feature domain is to compare the distance between the internal embedding vectors belonging to the same feature and the distance between the external vectors that do not belong to the same feature. The smaller the calculated distance, the more similar the two entities are, and vice versa. The comparison results are shown in Figure 15.

The blue histogram represents the average value of the distance calculation within the entity attribute domain, and the orange is the average value between the attribute domains. After calculating the average distance of each label and comparing them, it could be intuitively analyzed from the graph that the average distance between types, actors, countries, and entities within the label was significantly smaller than the external distance. The model could effectively learn the attributes of entity attributes. Entities with similar labels were closer in the space to express attribute information.

The experiment used Node2vec, GraphSage, Fast Random Projection, and hash2vec four-node entity embedding algorithms to compare and test the algorithm in this paper.

Table 5 shows the five closest movie entities in space computed by each algorithm.

The vector of embedded nodes obtained by the FADH algorithm had a similar expression effect to the existing graph node embedding algorithm under the constraints of entity attribute information rules. For example, the embedding vector obtained by the embedding of the FADH algorithm in the movie "Titanic" had the same number of 95% as the 20 nearest neighbors, calculated by the Node2vec algorithm, and 80% of the GraphSage algorithm and the FastRP algorithm. To sum up, the FADH algorithm in this paper had a better language expression effect.

In the experiment, the time and memory used to train the FADH and Node2vec models using the same data set are shown in Table 6.

An essential factor for the FADH algorithm to dynamically add data is that when an entity generates an embedded vector, it is only related to its attributes and does not involve other entities. Only its attributes can generate the corresponding vector representation. The training time only involves the embedding dimension. When the number of texts is fixed, the training time and memory are linearly related to the embedding dimension. The larger the embedding dimension, the longer the time and the larger the memory usage. Compared with other models that require data embedding and overall data training, the FADH model has significant advantages.

## 5. Conclusions

Based on deep neural networks and hashing algorithms, this paper proposes a novel embedding algorithm that can learn the attribute association between entities. An entity coding method that integrates attribute information is adopted to extract entity information from three aspects: coding fusion, feature extraction, and feature learning of entity attributes. To solve the problem whereby different entities may have the same embedding representation, this paper proposes using multiple hash functions to extract features to represent an entity. This effectively alleviates the problem of the model not being able to distinguish between different feature values, and thereby improves the model’s efficiency at the same time. This paper took film data as an example so as to introduce the coding method and specific algorithm process in detail and to realize the effect of dynamically adding data model reuse. The experimental setup compared the FADH algorithm designed in this paper with the Node2vec algorithm, GraphSage algorithm, and Fast Random Projection algorithm, respectively, and analyzed the time performance, memory consumption, and expression ability of different algorithms in embedding vectors of different dimensions. The experimental results highlighted the following: First, according to the five relevant entity properties closest in the vector space, referring to the first 20 nearest neighbors of the embedding vector under different algorithms, the similarity between the FADH algorithm and the GraphSage algorithm and the FastRP algorithm in this paper reached more than 80%, and the similarity with the Node2vec algorithm reached more than 95%; Second, compared with other algorithms, the FADH algorithm designed in this paper consumed the least time and occupied the smallest memory; Third, in summary, the FADH algorithm developed in this paper had a good language expression effect, and the performance of time and spatial complexity was significantly improved.

The FPDH algorithm designed in this paper has great potential in processing entity attribute information fusion. Next, we will pay more attention to the evaluation criteria that are more in line with the real situation, taking into account the impact of different characteristics on different entities.

In the future, it will still be a challenge to popularize the embedded algorithm with additional information in the personalized recommendation field of recommendation system research. At the same time, the impact weight between different features and different entities is key to content to be studied next. We still have a long way to go in applying the algorithm to more novel and meaningful applications.

## Figures and Tables

**Figure 1 entropy-25-00361-f001:**
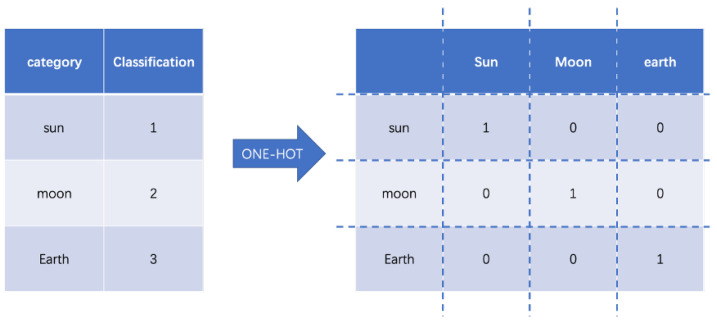
One-hot encoding example.

**Figure 2 entropy-25-00361-f002:**
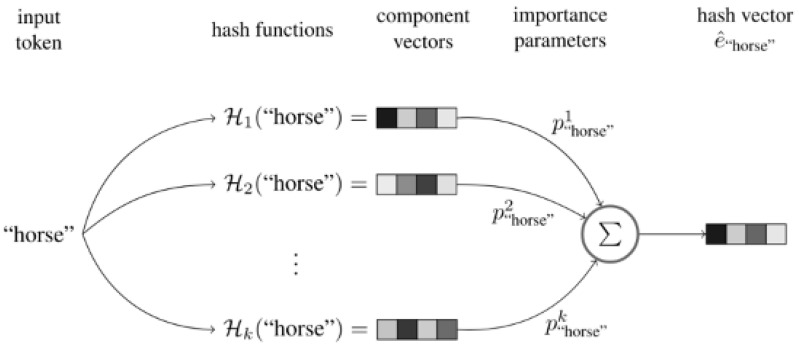
Hash embedding.

**Figure 3 entropy-25-00361-f003:**
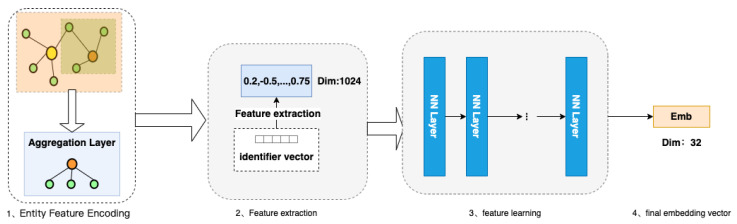
FADH algorithm model.

**Figure 4 entropy-25-00361-f004:**
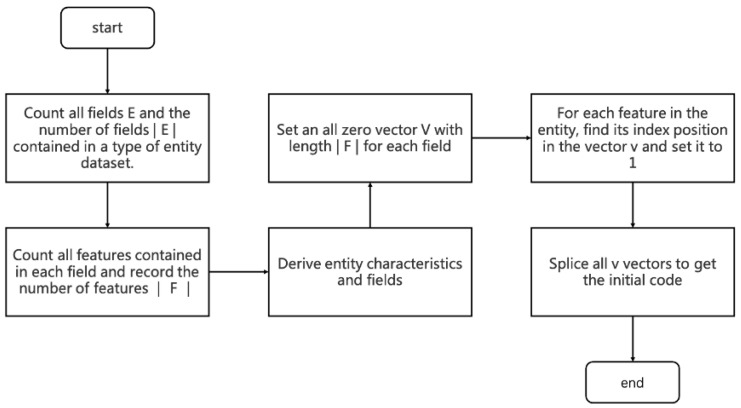
Coding Flowchart.

**Figure 5 entropy-25-00361-f005:**
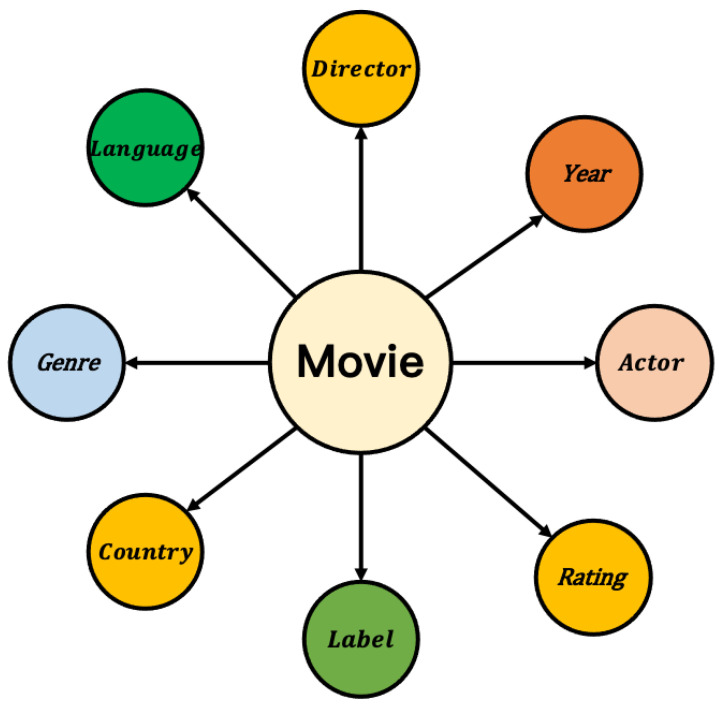
Domains contained in the movie entity.

**Figure 6 entropy-25-00361-f006:**
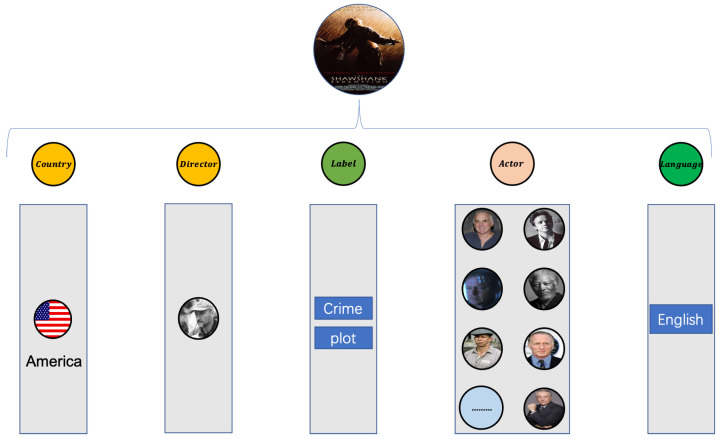
The movie example of The Shawshank Redemption.

**Figure 7 entropy-25-00361-f007:**
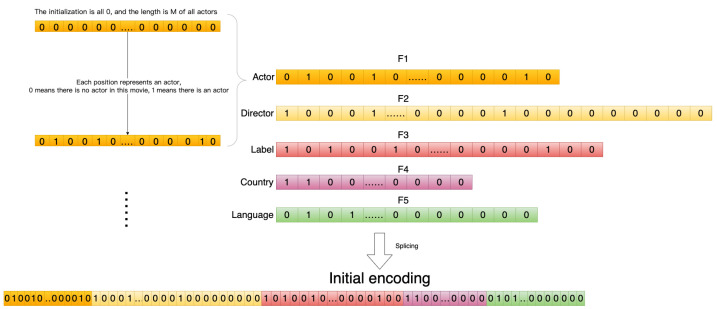
Initial encoding.

**Figure 8 entropy-25-00361-f008:**
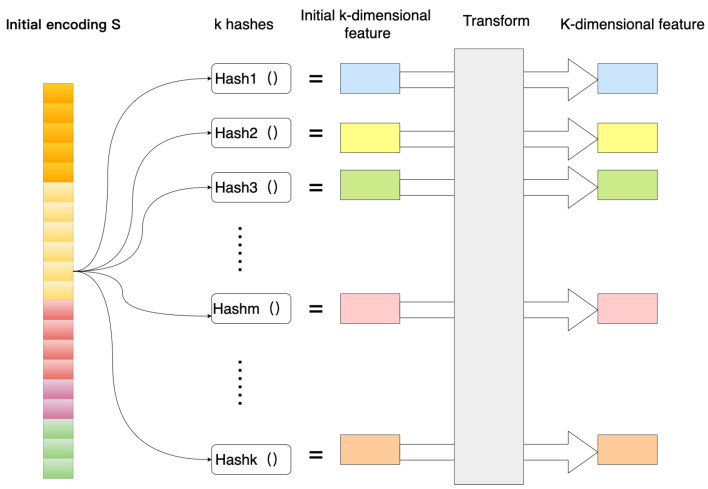
Multiple hash encoding.

**Figure 9 entropy-25-00361-f009:**
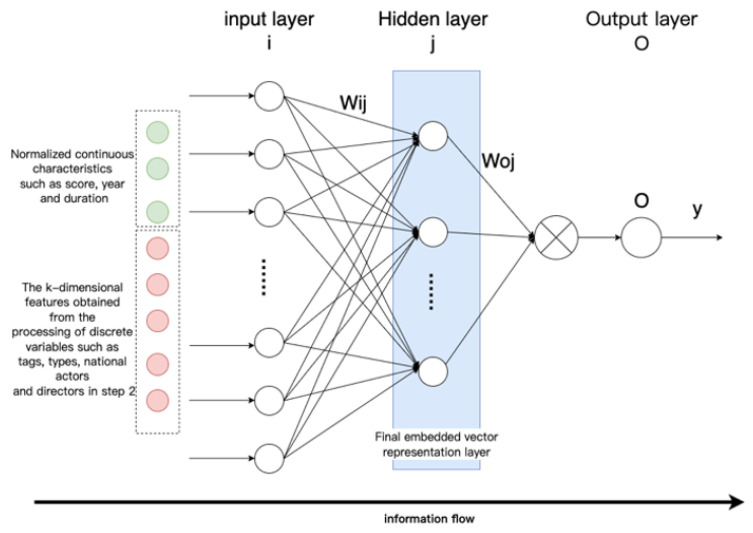
Schematic diagram of feedback neural network feature learning.

**Figure 10 entropy-25-00361-f010:**
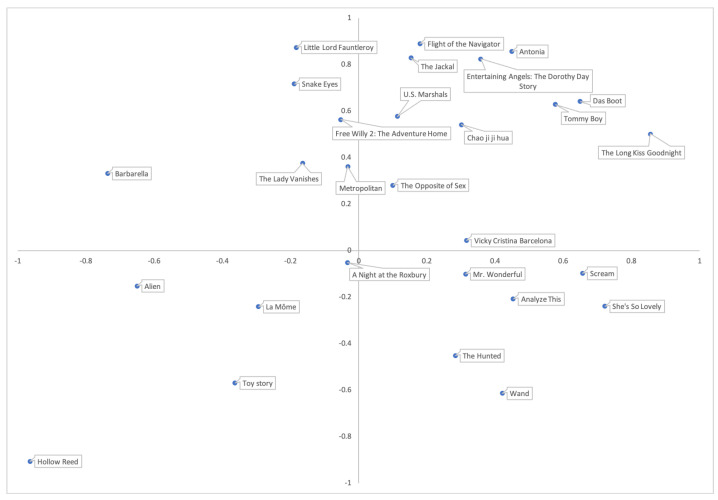
Dimensionality reduction visualization.

**Figure 11 entropy-25-00361-f011:**
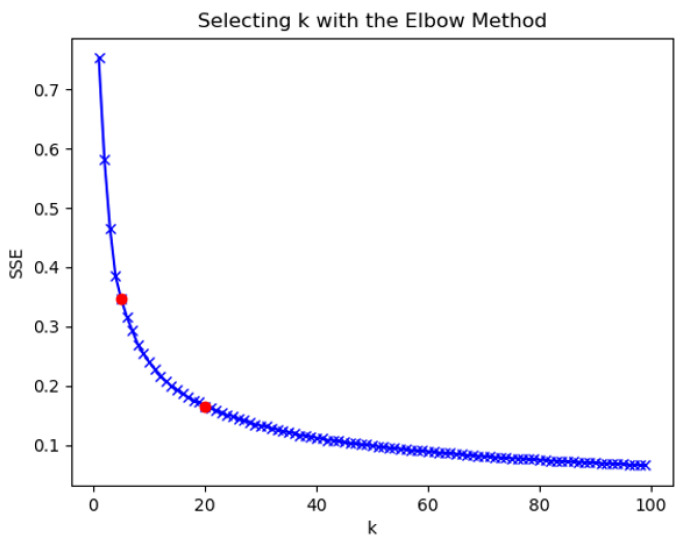
Clustering coefficient.

**Figure 12 entropy-25-00361-f012:**
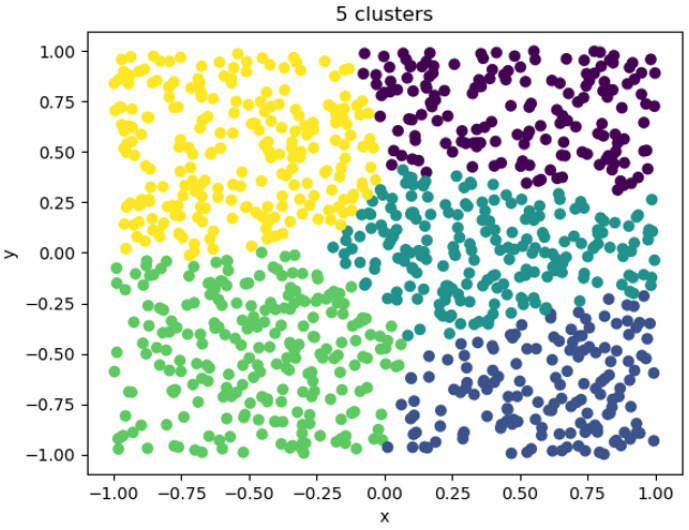
Five clusters visualizations.

**Figure 13 entropy-25-00361-f013:**
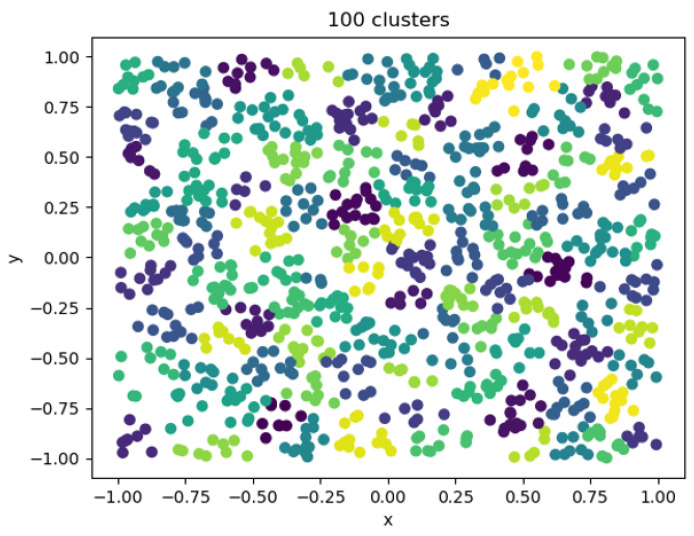
One hundred cluster visualizations.

**Figure 14 entropy-25-00361-f014:**
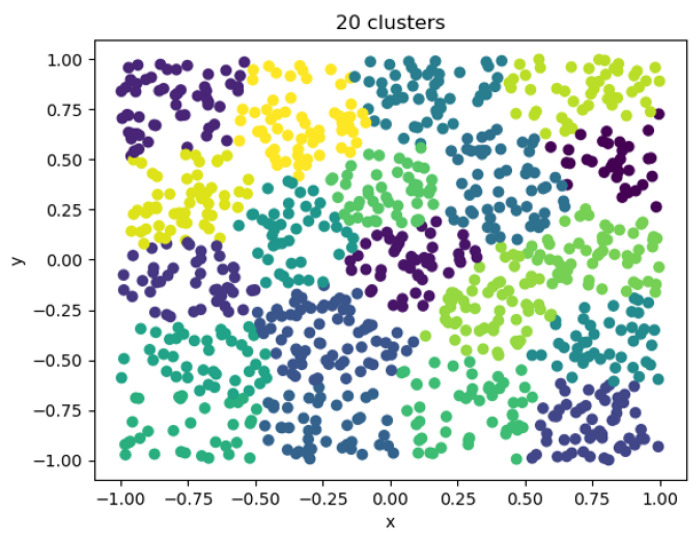
Twenty cluster visualizations.

**Figure 15 entropy-25-00361-f015:**
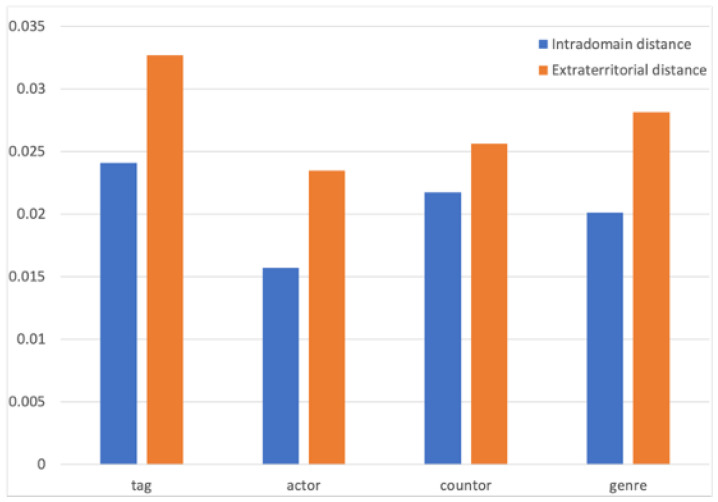
Intra-domain distance comparison.

**Table 1 entropy-25-00361-t001:** Symbol Description.

Symbol Definition	Meaning Description
s	Feature word
e	Embedding vector
V	Feature value (representation of feature word)
W	Embedding table: one-hot coding with all feature words
N	number of datasets
k ∈N	number of hash functions
n ∈N	the size of the vocabulary
d ∈N	Embedded dimension
m ∈N	Size of the hash table (usually m < n)
H: V → [m]	Hash function: map feature words to 1, 2, 3, …, m

**Table 2 entropy-25-00361-t002:** Dataset statistics.

Dataset	Total	Genre	Actor	User	Ratings
**hetrec2011-movielens-2k-v2**	10,197	20	95,321	2113	855,598

**Table 3 entropy-25-00361-t003:** Fully connected neural network structure.

	Input	Hidden Layer 1	Hidden Layer 2	Hidden Layer 3	Hidden Layer 4	Hidden Layer 5	Hidden Layer 6	Output Layer
1	1024	1024	1024	1024	32	32	32	1
2	1024	1024	1024	1024	64	64	64	1
3	1024	1024	1024	1024	128	128	128	1
4	1024	1024	1024	1024	256	256	256	1
5	1024	1024	1024	1024	1024	1024	1024	1

**Table 4 entropy-25-00361-t004:** Two-dimensional feature data after reduction by T-sne.

Movie Title	2-D Vector	
Scream	0.21266807615756989	0.5888321399688721
Tommy Boy	0.9140979647636414	0.7566967606544495
Alien	0.9574490785598755	0.41326817870140076
……		
She’s So Lovely	0.7865368723869324	0.10100453346967697
U.S. Marshals	0.0719037652015686	0.07177785784006119
……		
A Night at the Roxbury	0.5604234933853149	0.8568414449691772

**Table 5 entropy-25-00361-t005:** Top 5 nearest neighbor list.

	Node2vec	GraphSage	FastRP	FADH
Titanic	Revolutionary Road	Top Gun	Ladder 49	Ghost
	Top Gun	Casablanca	Mighty Joe Young	Revolutionary Road
	Ghost	Revolutionary Road	Revolutionary Road	Ladder 49
	Firestorm	Ghost	Casablanca	Firestorm
	Ladder 49	Firestorm	Top Gun	Top Gun
The Shawshank Redemption	The Green Mile	Fight Club	Fight Club	A Clockwork Orange
	One Flew Over the Cuckoo’s Nest	The Silence of the Lambs	One Flew Over the Cuckoo’s Nest	The Silence of the Lambs
	Fight Club	The Sixth Sense	The Green Mile	The Green Mile
	A Clockwork Orange	Mr. Smith Goes to Washington	The Sixth Sense	Fight Club
	Forrest Gump	The Green Mile	The Bridge on the River Kwai	One Flew Over the Cuckoo’s Nest

**Table 6 entropy-25-00361-t006:** Time and memory consumed by the model.

Dimension		32	64	128	256	1024
FADH	time/s memory/MB	10,431 6.3	10,600 11.2	10,833 21.1	13,092 40.6	40,319 158.4
Node2vec	time/s memory/MB	14,773 8.2	17,483 13.4	20,849 24.5	32,531 51.1	74,773 169

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
