# Peer review of "Design and Application of Deep Hash Embedding Algorithm with Fusion Entity Attribute Information"

_entropy, 2023, doi:10.3390/e25020361_

Round 1

Reviewer 1 Report

The paper has the potential to be accepted for publication. Before that, the authors are advised to consider the following comments and suggestions. Therefore, I recommend a minor revision for this submission.

1. Fig. 5 and Fig. 6 show highly similar content and are poorly presented, and the authors should redraw the figure.

2. On page 12, Fig. 10 is poorly presented, which should make it easy for the reader to see and understand what the figure is trying to express. The authors should redraw the figure and add more explanation about this result.

3. On page 13, the authors introduce three different phases of the clustering coefficient. But Fig. 12 only shows the result of the clustering coefficient is 5 to 13. It should include two extreme cases showing what happens when the clustering coefficient is extremely large or small. Besides, the presentation of 5 to 13 can be reduced.

4. Heading of 3.5 and 3.6 is the same. What's the difference between these two parts? The authors should explain the differences and use different headings.

5. There are some typos in this paper, for example on 370 and 371,'' at the samently time''.

Author Response

Dear editor

Thank you for processing the manuscript again. We also thank the reviewers for their efforts and time in reviewing our work. In response to the comments made by the reviewers, we have revised the manuscript. The detailed point-to-point response is shown below.

Best regards

  • Reply to Reviewer 1

The paper has the potential to be accepted for publication. Before that, the authors are advised to consider the following comments and suggestions. Therefore, I recommend a minor revision for this submission.

First of all, thank you for your affirmation of this article, and then respond to each point.。

  1. 5 and Fig. 6 show highly similar content and are poorly presented, and the authors should redraw the figure.

Reply: Fig. 5 and Fig. 6 have been redrawn and clearly distinguished。

  1. On page 12, Fig. 10 is poorly presented, which should make it easy for the reader to see and understand what the figure is trying to express. The authors should redraw the figure and add more explanation about this result.

Reply: Fig 10 has been redrawn and explanations have been added,For example, an explanation of the similarity of the physical attributes of the two films "Das Boot" and "tommy Boy" has been added in the article。

  1. On page 13, the authors introduce three different phases of the clustering coefficient. But Fig. 12 only shows the result of the clustering coefficient is 5 to 13. It should include two extreme cases showing what happens when the clustering coefficient is extremely large or small. Besides, the presentation of 5 to 13 can be reduced.

Reply:This part has been revised to show the two extreme cases of clustering coefficient at 5 and 100, and more explanations have been added. In addition, when the clustering coefficient is less than 5, the number of categories is too small to have practical significance。

  1. Heading of 3.5 and 3.6 is the same. What's the difference between these two parts? The authors should explain the differences and use different headings.

Reply:Thank the judges for their careful review. Different titles have been used in the revised version。

  1. There are some typos in this paper, for example on 370 and 371,'' at the samently time''.

Reply:Grammar problems and spelling errors have been corrected throughout the article。

Reviewer 2 Report

The paper entitled "Design and Application of Deep Hash Embedding Algorithm with Fusion Entity Attribute Information(FADH)" is well written. Presented research results are up to the academic standards. The paper has a lot of merit and should be recommended for publication, but after correcting some important issues from the point of view of the quality of "Entropy" journal.

1. I would like to suggest authors to extension of the introduction so that this section can fully present the current state of art.

2. I also propose to add nomenclature (description of used symbols and abbreviations) to improve the readability of the article.

3. Figures presented in the paper are not readable and make it difficult for the reader to precisely analyze the presented research results.

4. Although the topic and research results presented in the paper is current and interesting, the references section in the peer-revied paper is poor. The references section does not contain the current state of knowledge in the scope presented in the peer-revied paper.

5. In equations (12) and (14), scalar multiplication should be used instead of the convolution symbol.

6. Comparing algorithm runtimes is unreliable. The time (apart from the hardware) is influenced by the preparation of the implementation itself (optimal operation of the code). I suggest assessing the computational complexity of algorithms using the Big O metric.

7. The methodology of research description needs significant improvement. Verification methods should be clearly characterized in order to properly assess the performance of individual algorithms.

Author Response

Dear editor

Thank you for processing the manuscript again. We also thank the reviewers for their efforts and time in reviewing our work. In response to the comments made by the reviewers, we have revised the manuscript. The detailed point-to-point response is shown below.

Best regards

The paper entitled "Design and Application of Deep Hash Embedding Algorithm with Fusion Entity Attribute Information(FADH)" is well written. Presented research results are up to the academic standards. The paper has a lot of merit and should be recommended for publication, but after correcting some important issues from the point of view of the quality of "Entropy" journal.

First of all, thank you for your affirmation of this article, and here we reply to your valuable suggestions one by one.

  1. I would like to suggest authors to extension of the introduction so that this section can fully present the current state of art.

Reply:The introduction part of this article has been modified in detail, and a more detailed description has been added to the relevant algorithms and background algorithms mentioned. In addition, the depth and breadth of the introduction are also expanded.

  1. I also propose to add nomenclature (description of used symbols and abbreviations) to improve the readability of the article.

Reply:Relevant symbols and abbreviations mentioned in the text have been added in detail.

  1. Figures presented in the paper are not readable and make it difficult for the reader to precisely analyze the presented research results.

Reply:The numbers in the article are used as the display, and different colors are used to distinguish the size relationship between the data, and the meaning of the representation is explained.

  1. Although the topic and research results presented in the paper is current and interesting, the references section in the peer-revied paper is poor. The references section does not contain the current state of knowledge in the scope presented in the peer-revied paper.

Reply:This revision also cites high-quality references。

  1. In equations (12) and (14), scalar multiplication should be used instead of the convolution symbol.

Reply:The symbol problem in the formula has been modified。

  1. Comparing algorithm runtimes is unreliable. The time (apart from the hardware) is influenced by the preparation of the implementation itself (optimal operation of the code). I suggest assessing the computational complexity of algorithms using the Big O metric.

Reply:Due to the use of deep neural network and other deep learning models in the article, it will be difficult to determine whether the model converges and the training rounds. It is more difficult to calculate the time complexity with Big O metric, which is judged by experience. The algorithm proposed in this paper combines the advantages of fast computation of hash algorithm in theory. Intuitively, the overall operation result should be faster. The algorithm used in comparison achieves the same effect on the same computer for time comparison. Through simple examples, the previous conjecture can be indirectly proved

  1. The methodology of research description needs significant improvement. Verification methods should be clearly characterized in order to properly assess the performance of individual algorithms.

Reply:In the revised version, the description method of the article is improved, and the verification method also calculates the percentage of similarity with the benchmark model as the evaluation index.

Reviewer 3 Report

This work has taken a snippet of artificial intelligence algorithms, how the FADH process can be applied to movie databases
with time and space optimization as the ultimate goal. The basic question may be, what is the point of this?
Many highly relevant cancer research databases or self-driving car databases struggle with terabytes of data to cluster and optimum search.
That the movie database is here the example is beyond comprehension.

The paper is difficult to read, difficult to understand, unclear, full of typographical errors, the figures are illegible and unreadable,
the figure captions are unacceptable, i.e. the authors have not taken any care at all to make it understandable.

The reviewer gets the feeling that the authors may have followed the following train of thought: send in our half-finished work to the "Entropy" because of  time constraints, the reviewers will certainly read it carefully and, having corrected the errors listed, we have a publication.

The bibliography will only include papers of lower quality, such as conference papers or similar. IEEE publications are not included.

The paper lacks a link to GitHub with the source code for the algorithms. Without this, this article and any AI algorithm is not acceptable.

If it is the NLP model --> consequently the embedding scheme is used in science to characterize nonlinear systems in physics. (embedded dimension ans Lyapunov systems)
The difference or identity is not explained.

The work lacks an illustrative example, without it the definitions of entity, attribute etc. are not understandable.

However, the fundamental problem is: who will define and attirbute and the features of the entities of the systems to be modelled for hundreds of
thousands of instances? How can this be automated e.g. in a cancer research database?

Furthermore, how can I create a closed-loop system and decide if what I have developed is correct?

Going further, where can deep learning characteristics be found in feature learning. What is the reason for the 4 tables with 17 digit accuracy. (Table 4)
This is the rate used in relativistic quantum field theory at CERN in Geneva, Switzerland, to detect positrons during annihilation. Is that what we are talking about here?

I could list the serious errors at great length, but I propose a complete revision of the article, now based on medical databases and much more thoroughly.

Author Response

Dear editor

Thank you for processing the manuscript again. We also thank the reviewers for their efforts and time in reviewing our work. In response to the comments made by the reviewers, we have revised the manuscript. The detailed point-to-point response is shown below.

Best regards

Thank you for pointing out many problems and deficiencies in this article. I have also made a lot of changes to the content of the article. Here are some additional explanations:

  • The background of this article is also a part of the overall work, which is the research and implementation of recommendation algorithm based on movie database. The embedded coding mentioned in the article is a formal preliminary work. For example, Youtube's recommendation model (Deep Neural Networks for YouTube Recommendations) uses ID features, type features, and fixed-length movie name features to connect and add together to get the feature coding of the movie, and then calculates cosine similarity with the user's feature to get the recommendation list. Feature coding is an indispensable step, which is the significance of this article.
  • For the embedding method proposed in this paper, compare with the original data, and compare with the encoded data can effectively express the characteristics of the original data. For example, the original data has 20 types of movies. In the experiment, it is not known how many types are more appropriate to divide the encoded data into. However, based on the characteristics of the clustering algorithm, the final result obtained by determining the clustering coefficient by K-elbow method is more consistent with the actual situation, so the effectiveness of the method can be determined. The purpose of this paper is to propose a coding method that can reflect the characteristics of entity attributes.
  • This revised version has revised a large number of language descriptions, symbolic noun explanations, grammatical problems, and caption errors.
  • Github link will be provided in the modified version.

Round 2

Reviewer 2 Report

Authors considered or referred to all the comments indicated by me. The article can be published in its current form.

Reviewer 3 Report

My first thought is to acknowledge the authors' efforts (github, making some of the figures understandable).
However, I do not see a significant change in the paper in terms of comprehensibility and elaboration (theory, figures, bibliography, equations), and I still do not recommend the publication of the paper. The theory that has been developed cannot be interpreted without a serious background (IEEE Transactions on Neural Networks and Learning Systems) and requires a level of research that cannot be expected from the average reader. As it stands, the paper is not understandable.
The point of science is to build new solutions on existing solid, proven foundations. This is not the case at the moment.
I suggest that the authors write their paper on a completely new basis, with applications, with a focus on public understanding, and rewrite their paper after some time has passed, after they have presented the theory at a few conferences.